# The SUMO Family: Mechanisms and Implications in Thyroid Cancer Pathogenesis and Therapy

**DOI:** 10.3390/biomedicines12102408

**Published:** 2024-10-21

**Authors:** Bahejuan Jiaerken, Wei Liu, Jiaojiao Zheng, Weifeng Qu, Qiao Wu, Zhilong Ai

**Affiliations:** 1Department of Surgery (Thyroid & Breast), Zhongshan Hospital, Fudan University, Shanghai 200032, China; 2School of Basic Medical Sciences, Fudan University, Shanghai 200032, China

**Keywords:** SUMOylation, thyroid cancer, therapeutic targets, SUMO inhibitor, TAK-981, drug sensitivity

## Abstract

(1) Background: Small ubiquitin-like modifiers (SUMOs) are pivotal in post-translational modifications, influencing various cellular processes, such as protein localization, stability, and genome integrity. (2) Methods: This review explores the SUMO family, including its isoforms and catalytic cycle, highlighting their significance in regulating key biological functions in thyroid cancer. We discuss the multifaceted roles of SUMOylation in DNA repair mechanisms, protein stability, and the modulation of receptor activities, particularly in the context of thyroid cancer. (3) Results: The aberrant SUMOylation machinery contributes to tumorigenesis through altered gene expression and immune evasion mechanisms. Furthermore, we examine the therapeutic potential of targeting SUMOylation pathways in thyroid cancer treatment, emphasizing the need for further research to develop effective SUMOylation inhibitors. (4) Conclusions: By understanding the intricate roles of SUMOylation in cancer biology, we can pave the way for innovative therapeutic strategies to improve outcomes for patients with advanced tumors.

## 1. Introduction

With over 821,000 cases worldwide in 2022 [1], thyroid cancer ranks as the seventh most common cancer in terms of overall incidence and fifth in women. The incidence rate is three times higher in women than in men. China alone accounted for over half of the global incidence burden, with 466,000 new cases in 2022. Thyroid cancer is classified into three main histological types: (1) differentiated thyroid cancer (DTC), which includes papillary, follicular, and oncocytic carcinomas; (2) medullary thyroid cancer (MTC), sometimes linked with multiple endocrine neoplasia type 2 syndromes; and (3) anaplastic thyroid cancer (ATC), often evolving from differentiated thyroid cancer and associated with high mortality [2]. A study across 25 countries found that the increase in thyroid cancer was primarily confined to papillary carcinomas, which are often detected through intensive thyroid gland screening [3]. For many years, the primary approach to treating thyroid cancer has been the surgical removal of the thyroid gland.

In some cases, particularly for DTC, this surgery is followed by post-operative treatment with radioactive iodine (RAI) and suppressive doses of thyroid replacement hormones. Surgery often results in a cure for most patients with well-differentiated thyroid cancer. Additionally, RAI therapy after surgery has been shown to enhance overall survival in patients who are at a high risk of recurrence. In the management of metastatic thyroid cancer, there is a growing use of antiangiogenic multikinase inhibitors and targeted therapies aimed at specific genetic mutations responsible for the cancer [4].

Protein post-translational modification (PTM) by small ubiquitin-like modifiers (SUMOs) involves the wide-spread regulation of many cellular functions inside the cell of eukaryotic organisms. The SUMO protein is a member of the ubiquitin-like modifier (UbL) protein superfamily: These proteins are composed of analogous small protein domains that are conjugated to target proteins by creating an isopeptidic linkage between the lysine groups of the substrate and the C-terminal end of the UbL protein. This linkage is characterized by the presence of a distinctive diglycine motif at the C-terminus of the UbL protein [5,6,7,8,9]. Since their initial identification in 1996 [10], SUMO proteins have been found to exist in five isoforms in mammalian cells: SUMO1, SUMO2, SUMO3, SUMO4, and SUMO5. While SUMO1–3 are broadly expressed across human tissues, SUMO4 and SUMO5 exhibit more restricted expression patterns, particularly in the testis and lymphocytes [11]. The dynamic nature of SUMO modification (SUMOylation), a reversible modification process, plays a critical role in modulating protein localization, stability, and function, thereby influencing key biological pathways such as DNA repair, transcriptional regulation, and cellular stress responses [12,13]. Recent studies have highlighted the multifaceted roles of SUMOylation in cancer biology, particularly in thyroid cancer, where aberrant SUMOylation patterns contribute to tumorigenesis and progression [14,15]. This review explores the intricate mechanisms of SUMOylation and its implications in thyroid cancer, emphasizing potential therapeutic strategies that target the SUMOylation pathway to improve patient outcomes.

## 2. SUMO Proteins and SUMO Catalytic Cycle

### 2.1. SUMO Family in Human

Essential for cellular homeostasis, SUMO proteins regulate various biological processes via a dynamic SUMOylation–deSUMOylation cycle. SUMO proteins, first discovered in 1996, play a pivotal role in PTM in eukaryotic cells, with five isoforms—SUMO1–5—that structurally resemble ubiquitin [16].

SUMO proteins, which come in five isoforms (SUMO1, SUMO2, SUMO3, SUMO4, and SUMO5), are crucial in post-translational modification across human tissues. While SUMO1–3 are ubiquitously expressed, SUMO4 and SUMO5 follow tissue-specific patterns, with SUMO5 being particularly abundant in testis and blood lymphocytes [11]. SUMO2 and SUMO3, due to their high amino acid sequence similarity, are collectively referred to as SUMO-2/3 [17,18]. SUMO1 typically modifies proteins related to physiological status, while SUMO-2/3 is involved in stress-responsive protein modifications [19]. It is hypothesized that SUMO-2/3 may compensate for SUMO1 in its typical protein targets [20]. Current research on SUMO4 and SUMO5 suggests potential associations with diabetes and leukemia, respectively, though further investigation is ongoing [21].

### 2.2. SUMO Catalytic Cycle

SUMO proteins, initially present as inactive precursors with a molecular weight of around 11 kDa, undergo activation via proteolytic cleavage catalyzed by enzymes such as ULP1 in yeast or Sentrin/SUMO-specific protease 1 (SENP1) in humans. This cleavage exposes a crucial diglycine motif for subsequent interactions [22,23]. SUMO activation is mediated by the heterodimeric E1-activating enzyme (SAE1/SAE2), which forms a thioester bond between the SUMO and its internal cysteine residue (Cys593) through ATP hydrolysis [24,25,26]. Subsequently, the activated SUMO is transferred to the cysteine residue at position 93 of UBC9, the sole E2-conjugating enzyme, via a transesterification reaction, generating an E2-SUMO thioester [27,28,29,30,31,32]. UBC9 catalyzes the conjugation of SUMO to lysine residues of target proteins, completing the SUMOylation process. This process is facilitated by E3 ligases, which stabilize the E2-SUMO thioester conformation, enabling its conjugation to the substrate lysine residue within a SUMO-interacting motif (SIM) [33,34,35]. This multi-step enzymatic cascade ensures the accurate and efficient modification of target proteins by the SUMO [36,37,38].

DeSUMOylation, a reversible and dynamic process, is terminated by SUMO proteases known as SENPs or Sentrin/SUMO-specific proteases, Figure 1. Six SENPs have been identified in humans, with unique cellular locations and substrate specificities [39]. SENP1 and SENP2, localized primarily at the nuclear pore, can process all three SUMO isoforms (SUMO1, 2, and 3) and remove both mono- and polymeric SUMOylated proteins. By contrast, SENP3 to SENP7 exclusively process SUMO-2/3, with SENP6 and SENP7 exhibiting sole hydrolase activity [40]. These proteases play a pivotal role in embryonic development, reflecting their distinct functions [41,42,43]. Recent discoveries include SUMO proteases deSUMOylating isopeptidase 1 (DESI1), DESI2, and USPL1, which have little sequence identity with the ULP/SENP family [44,45]. Each SENP exhibits various cellular locations and substrate specificities [39,46,47].

## 3. The Multifaceted Functions of SUMOylation in Protein Localization, Stability, and Genome Integrity

Over the past few years, SUMOylation has become a crucial regulator of diverse cellular processes, including protein localization, stability, and genome integrity, with important roles in transcriptional regulation.

### 3.1. SUMOylation in DNA End Resection and Genome Stability

Zhang et al. showed that MRE11 SUMOylation and ubiquitylation are dynamically controlled by PIAS1 and SENP3 to facilitate DNA end resection and genome stability [12]. Additionally, the SUMOylation of MORC2, a chromatin-remodeling enzyme, is crucial for chromatin remodeling and DNA repair in response to DNA damage [13]. These findings underscore the importance of post-translational modifications, particularly SUMOylation, in regulating DNA repair mechanisms and genome stability.

### 3.2. Role of SUMOylation in Maintaining Protein Stability

The dysregulation of the SUMOylation process can lead to the loss of HNF4α and hepatic function, underscoring its vital role in maintaining the hepatocellular phenotype. It has been established that the SUMOylation of HNF4α regulates its protein stability and potentially its transcriptional activity [48]. Furthermore, the pathogenic mutations in the TRAIP gene are associated with primordial dwarfism in patients [49]. SUMOylation has been demonstrated as critical for ensuring the proper subcellular localization and protein stability of TRAIP, which exhibits various functions in the nucleus [50]. Notably, Hamard, P.J., et al. discovered that the ATF7 transcription factor undergoes SUMOylation both in vitro and in vivo. SUMOylation affects the intranuclear localization and transcriptional activity of the ATF7 transcription factor by interfering with its interaction with TAF12, thus impeding its access to specific promoters [51]. These findings underscore the significance of SUMOylation in various biological processes.

### 3.3. Impact of SUMOylation on Receptors

In the intricate landscape of cellular regulation, SUMOylation emerges as a crucial mechanism modulating the activities of various proteins. Among these, peroxisome-proliferator-activated receptor γ (PPARγ) and pregnane X receptor (PXR) are pivotal players in lipid metabolism and xenobiotic responses, respectively.

PPARγ, a ligand-activated nuclear receptor regulating sugar and lipid metabolism, can be SUMOylated to modulate its activity [52]. The SUMOylation of PPARγ can inhibit its activity, thus affecting lipid metabolism. In lung cancer cells, SUMO modification of PPARγ induces lipid-metabolism-related gene expression, promoting lipid synthesis and NADPH consumption. This process enhances β-oxidation and mitochondrial reactive oxygen species (ROS) production, leading to tumor suppression [53].

In the liver, SUMOylation and ubiquitination of the nuclear receptor pregnane X receptor (PXR/NR1I2) regulate its biological functions, particularly in response to xenobiotic or inflammatory stimuli. The nuclear receptor PXR/NR1I2 is a key regulator in xenobiotic responses, involved in the metabolism and clearance of toxic substances, as well as inflammatory reactions. Specifically, ubiquitination promotes inflammatory responses, while SUMOylation inhibits them [54].

### 3.4. Regulatory Role of SUMOylation in Ras Proteins

Recent research has shed light on the regulatory impact of Ras proteins via SUMOylation, a process in which all three isoforms of Ras proteins (HRas, KRas, and NRas) are modified by SUMO3 [55]. Identifying lysine 42 as the key site responsible for this modification is of particular significance. Studies involving the KRas V12/r42 mutant discovered that the mutation impedes the activation of the Raf/MEK/ERK signaling pathway, resulting in a decrease in cell migration and invasion rates in diverse in vitro cell models [56]. Additionally, blocking SUMO E2 in pancreatic cells undergoing transformation was found to reduce cell migration dependent on the dosage, corresponding to diminished levels of KRas SUMOylation and the expression of mesenchymal cell markers. Further evidence from experiments using mice as model organisms has shown that introducing SUMO-resistant mutants can impede tumor growth in vivo. Collectively, these findings support the idea that SUMOylation plays a vital role in regulating the functions of Ras in processes such as cell proliferation, differentiation, and malignant transformation. Thus, targeting the SUMO modification system of Ras oncoproteins may offer a promising and innovative therapeutic strategy for addressing various human malignancies.

## 4. SUMOylation and Thyroid Cancer

### 4.1. CCDC6 and SUMOylation

Cyclic AMP (cAMP)-response-element-binding protein 1 (CREB) is a 43 kDa stimulus-induced transcription factor (TF) [57]. The overexpression of CREB is associated with aberrant signal transduction caused by the deregulated expression of downstream genes that control the hallmarks of cancer, such as proliferation, apoptosis, angiogenesis, metastasis, immune surveillance, and metabolism, and the generation of tumor stem cells, which lead to the initiation and progression of tumors. These different CREB activities result in increased tumor growth, resistance to antiproliferative signals, decreased apoptosis, enhanced angiogenesis, increased metabolism, and reduced immunogenicity.

The RET/papillary thyroid carcinoma 1 (PTC1) oncogene, frequently found in human papillary thyroid carcinomas, involves the fusion of RET’s kinase domain with the initial 101 amino acids of CCDC6, leading to allelic expression loss and influencing thyroid cancer development [58]. This fusion reduces the CCDC6-mediated inhibition of CREB1, resulting in increased CREB1 activity and the upregulation of its target genes, such as AREG and cyclin A, thereby promoting thyroid tumorigenesis [59]. CCDC6, a tumor repressor known for its pro-apoptotic effects [60,61], undergoes SUMOylation, influencing its tumor-suppressive functions. SUMOylation leads to the cytoplasmic sequestration of CCDC6 and a decrease in its interaction with CREB1 (Figure 2), thereby promoting CREB1-dependent transcriptional activity and cellular proliferation [14]. This mechanism highlights the dual role of SUMOylation in regulating tumor suppressor activity and promoting thyroid cancer progression (Figure 2).

### 4.2. PDGF-C and SUMOylation

Platelet-derived growth factor-C (PDGF-C), a key growth factor in cancer progression, plays a crucial role in promoting growth, angiogenesis, and tumorigenesis in various types of cancers [62,63,64]. Recent studies have shown that in thyroid cancer cells, the levels of SUMOylated PDGF-C in the nucleus are significantly lower than normal thyroid cells [65] (Figure 2). This decrease in SUMOylation may play a role in the development of thyroid cancer by impacting growth, angiogenesis, and tumor formation. However, further research is needed to fully understand the mechanisms involved, whether through inhibited SUMOylation or increased deSUMOylation.

### 4.3. TFAP2A and SUMOylation

Recent findings indicate that the progression from papillary to anaplastic thyroid cancer in cell models may be driven by the SUMOylation of transcription factor TFAP2A, which alters gene expression patterns linked to anaplastic thyroid cancer [15]. Follow-up studies using SUMO inhibitors, PYR-41 and anacardic acid, in murine models of anaplastic thyroid cancer demonstrated reductions in tumor size and enhanced tumor-free survival (Figure 2), suggesting that targeting this post-translational modification could potentially ameliorate outcomes in anaplastic thyroid cancer. However, these promising outcomes from cell and animal models need further validation and cautious interpretation before clinical application.

### 4.4. Deregulation of SUMOylation Machinery in Thyroid Cancer

A comprehensive analysis of the expression of SUMOylation machinery components in papillary thyroid cancer (PTC) reveals the significant deregulation of SENP8, ZMIZ1, SAE1, PIAS1, and PIAS2 in most cases [66] (Figure 2). Although these alterations do not correlate with clinicopathological parameters, they likely contribute to the PTC phenotype, underscoring the complex role of SUMOylation in thyroid cancer pathogenesis.

### 4.5. The Role of PIAS2b in Anaplastic Thyroid Carcinomas

The PIAS family (PIAS1-4) comprises nuclear, zinc-binding proteins distinguished by a Siz/PIAS (SP)-RING domain that functions as an E3 SUMO ligase [67]. Among these, PIAS2 is highly expressed in differentiated papillary thyroid carcinomas but significantly reduced in anaplastic thyroid carcinomas (ATC), a highly lethal, undifferentiated cancer. Recent research [68] identified PIAS2b as essential for mitosis in ATC cells. Silencing PIAS2b with dsRNAi selectively induces cell death in these aggressive cancer cells by disrupting spindle assembly, impairing chromosome–microtubule attachment, and enhancing proteasome activity (Figure 2). This silencing leads to reduced levels and the SUMOylation of key mitotic proteins (e.g., Tubulin gamma, PLK1, CDK1, PSMC5, TUBB3, and PPP2CA), culminating in mitotic catastrophe. Notably, PIAS2b-dsRNAi specifically targets anaplastic cancer cells, both thyroid and non-thyroid, while sparing normal or hyperplastic cells, underscoring its potential as a therapeutic strategy for these aggressive cancers.

## 5. The Potential Application of SUMOylation in the Treatment of Thyroid Cancer

Thyroid cancer is a prevalent form of endocrine malignancy worldwide, showing an increased incidence rate in recent years. While conventional treatment options like surgery and radiation therapy are commonly used, some patients do not respond well to these methods or face recurrence and metastasis after initial treatment. Differentiated thyroid cancer (DTC) constitutes the majority of thyroid malignancies, representing roughly 80–90% of diagnosed cases [69]. This category predominantly includes papillary and follicular carcinomas. Total thyroidectomy (TT) and thyroid lobectomy (TL) are the main surgical approaches to DTC, with low to intermediate risk of recurrence. Complications arising from these procedures, notably injury to the recurrent laryngeal nerve and hypocalcemia due to parathyroid gland dysfunction, can significantly affect the patient’s overall well-being. In the pediatric population, as observed in adults, there has been a documented increase in the incidence of thyroid cancer over the past few decades [70]. The risk of surgical complications in children is elevated compared with the adult patient cohort. Given the exceedingly low disease-specific mortality rate in pediatric DTC patients, it is of the utmost importance to minimize the morbidity associated with treatment [71]. To enhance the effectiveness of thyroid cancer treatment, it is imperative to delve into innovative therapeutic approaches and molecular targets. The SUMOylation pathway, a crucial cellular modification process, has emerged as a promising target for thyroid cancer therapy. By focusing on this pathway, researchers aim to develop new strategies that may revolutionize the outcomes of thyroid cancer treatment and offer hope to patients battling this disease.

The dual roles of SUMOylation on substrates disrupt normal cellular processes, thereby playing a significant role in cancer promotion and suppression. On the one hand, SUMOylation primarily promotes oncogenic effects, driven by the deregulation of SUMO machinery components and the abnormal SUMOylation of key oncoproteins and tumor suppressors. Conversely, while SUMOylation does have tumor-suppressive effects, these are relatively minor, indicating a need for further research in this area. Thus, a comprehensive investigation into the targets and effects of SUMOylation will bolster confidence in the efficacy of SUMOylation-based cancer therapies. A growing body of research strongly supports the oncogenic roles of SUMOylation in tumor invasion, metastasis, angiogenesis, DNA damage and repair, and metabolic reprogramming. These oncogenic mechanisms offer potential targets and avenues for SUMOylation-based cancer therapies.

### 5.1. Potential Therapeutic Targets in the SUMOylation Pathway

#### 5.1.1. Targeting Tumor Invasion and Metastasis

The hallmark of tumor cell dissemination is the invasive metastatic cascade, the most lethal aspect of tumors. Notably, the SUMOylation cascade regulates tumor metastasis by promoting tumor angiogenesis and the epithelial–mesenchymal transition (EMT). Tumor angiogenesis refers to the forming of new blood vessels that supply tumors with essential oxygen and nutrients for growth and metastasis. Various angiogenic signaling pathways regulate this process. Although the role of SUMOylation in tumor angiogenesis remains unclear, evidence suggests its involvement. HEY1 (hairy/enhancer of split related with YRPW motif), a transcription factor from the basic helix–loop–helix family, is recognized as a key player in developmental angiogenesis. Researchers have found that SUMOylation facilitates the formation of the HEY1 transcriptional complex and enhances its DNA-binding capacity in endothelial cells. Consequently, SUMOylation preserves HEY1’s role as a repressive transcription factor that regulates numerous angiogenic genes, including receptor tyrosine kinases (RTKs) and components of the Notch pathway [72]. Another study involving HCC stem cells found that the deSUMOylation of hypoxia-inducible factor (HIF-1α) and Oct4 reduced their nuclear accumulation, thereby inhibiting tumor angiogenesis and maintaining stemness [73].

The EMT, characterized by enhanced invasiveness and metastatic potential, is regulated by SUMOylation across various types of cancers. For example, ginkgolic acid inhibits the proliferation, migration, and EMT of gastric cancer cells by blocking the SUMOylation of IGF-1R (insulin-like growth factor 1 receptor), which is significantly upregulated in these cells [74]. Conversely, SUMOylation can inhibit the EMT and tumor metastasis. Specifically, the SUMOylation of annexin A6 slows cell migration and tumor growth by suppressing the RHOU/AKT1-mediated EMT in hepatocellular carcinoma [75].

#### 5.1.2. Targeting the DNA Damage Response

The DNA damage response (DDR) is crucial for maintaining genomic stability. The inherent genomic instability of rapidly proliferating tumors presents therapeutic opportunities to target DDR pathways, enabling the selective destruction of cancer cells through additional replication stress, exogenous DNA damage, or DDR inhibition. Two primary pathways, non-homologous end joining (NHEJ) and homologous recombination (HR), are used by cells to repair the most severe form of DNA damage known as double-strand breaks (DSBs). Studies have shown that post-translational modifications of proteins play a critical role in regulating double-strand break repair. In the NHEJ repair pathway, the ubiquitin E3 ligase RNF168 acts as a key protein that responds promptly to DNA double-strand break damage. SENP1 has been identified as a specific deSUMOylase of RNF168 and is highly expressed in colorectal adenocarcinoma. SENP1 reduces the SUMOylation of RNF168 in response to DNA damage, limiting its recruitment to damaged DNA sites and enhancing repair efficiency, leading to cancer cell resistance against DNA-damaging agents [76]. Conversely, TIP60 is rapidly deSUMOylated by SENP3, facilitating its interaction with DNA-PKcs after irradiation, which promotes NHEJ repair. It is suggested that leaking SENP3 levels increase tumor cell sensitivity to various DNA damage treatments [77].

#### 5.1.3. Targeting RNA Transcription

Several transcription factors and co-transcriptional regulators have been reported as SUMOylated proteins. The relationship between SUMOylation and RNA transcription is primarily reflected in the regulatory effects of SUMOylation modifications on transcription factors and the functional impact on RNA polymerase. Understanding the impact of SUMOylation on RNA metabolism may yield new therapeutic strategies for cancer treatment. Since most SUMOylation substrate proteins are localized in the nucleus, SUMOylation mainly inhibits global transcription activity [78]. Specifically, DAXX, a key regulator of gene expression, interacts with core histones and various proteins to function as a transcriptional co-repressor or co-activator [79], influencing genes involved in cell death, survival, and tumorigenesis. Its expression is elevated in several cancers, including prostate [80], ovarian [81], and gastric cancer [82]. Notably, the SUMO1 modification of DAXX enhances its recruitment to PML-NBs and promotes apoptosis in cancer cells [83]. Transcription profiling indicates that SUMOylation represses global transcription by inhibiting transcriptional elongation. SUMOs and MYC exert opposing effects on global gene expression by modulating the dynamic processes of the SUMOylation and deSUMOylation of CDK9, the catalytic subunit of the P-TEFb kinase, which is crucial for effective transcriptional elongation. Specifically, the SUMOylation of CDK9 leads to transcriptional repression, while MYC enhances global transcription by counteracting CDK9 SUMOylation [84]. As an oncogene, MYC belongs to a superfamily of genes that encode frequently activated oncoproteins in human cancers [85]. Since MYC promotes gene expression by inhibiting CDK9 SUMOylation, targeting CDK9 SUMOylation may represent a viable therapeutic strategy, especially since there are currently no approved direct inhibitors of MYC.

#### 5.1.4. Targeting Immune Evasion

Many cancers evade the immune system through distinct immune evasion strategies [86]. Targeting the SUMOylation cascade inhibits tumor immune evasion by altering the tumor microenvironment and reconstituting immune surveillance. ROS, central factors in regulating the tumor microenvironment, also contribute to tumor immune evasion [87,88]. Cytotoxic T-cells (CTLs) are key players in cellular defense within the adaptive immune response. CTLs recognize foreign antigens processed and presented by the MHC class I (MHC-I) antigen processing and presentation machinery (APM) of target cells. The loss or down-regulation of the MHC-I APM is a common cause of primary and acquired resistance to cancer immunotherapies. The pharmacological inhibition of SUMOylation (SUMOi) not only drives the activation and IFN-γ secretion of CTLs but also amplifies the IFN-γ-induced restoration of tumor-intrinsic MHC-I suppression, thereby reconstituting immune surveillance [89]. In addition to immune cells, SUMOylation in tumor cells can also facilitate tumor immune evasion. Specifically, the SUMOylation of programmed cell death protein-1 ligand (PD-L1) by TRIM28, an E3 ubiquitin ligase and SUMO ligase, stabilizes PD-L1 by hampering its ubiquitination and enhancing its SUMOylation, leading to T-cell inactivation and immune evasion in gastric cancer [90]. Emerging data demonstrate that protein modification by SUMO represents a novel target for activating antitumor immunity. Combining tumor immunotherapy with SUMOylation inhibitors may provide a promising strategy for overcoming resistance to immunotherapy.

#### 5.1.5. Targeting Metabolic Reprogramming

Metabolic reprogramming is a hallmark of tumor cells, characterized by alterations in metabolic pathways during their proliferation and progression [91]. The SUMOylation cascade plays a crucial role in metabolic reprogramming, including the Warburg effect and fatty acid metabolism [92]. Therefore, targeting the SUMOylation cascade presents a promising strategy to suppress metabolic reprogramming, improve the tumor microenvironment, inhibit tumor growth, and enhance the sensitivity to antitumor drugs. The Warburg effect, also known as aerobic glycolysis, represents a typical abnormality in glucose metabolism within tumors and is regulated by SUMOylation. Specifically, the SUMO1-induced SUMOylation of PKM2, through binding to the SUMO-interacting motif site IKII265-268, promotes PKM2 dimerization and nuclear translocation, thereby facilitating glycolysis in hepatocellular carcinoma (HCC) [93]. However, the SUMOylation cascade can also inhibit glycolysis in tumors. Hexokinase 2 (HK2), the first rate-limiting enzyme of glycolysis, is SUMOylated at K315 and K492 in prostate cancer cells, which inhibits its binding to mitochondria and consequently reduces glycolysis in tumor cells [94]. In addition to aerobic glycolysis, SUMOylation may disrupt fatty acid metabolism in tumors. Specifically, SENP2, a deSUMOylating protease, enhances fatty acid degradation and consumption by increasing the expression of PPARγ, CPT1A, ACSL1, and CD36 through deSUMOylating SETDB1, thereby generating more energy to support esophageal squamous cell carcinoma (ESCC) proliferation [95].

### 5.2. Therapeutic Potential and Development of SUMOylation Inhibitors

The E1 enzyme plays a crucial role in the SUMOylation process. It is emerging as a promising therapeutic target due to its involvement in essential biological functions that support cancer cell survival and progression, such as proliferative signaling, cell cycle regulation, and DNA damage response [96]. Preclinical studies have demonstrated that inhibiting E1 enzymes can restrict tumor growth and enhance antitumor immune responses, suggesting that targeting E1 enzymes represents a viable and rational approach for cancer therapy [97]. Inhibition of SUMOylation has been shown to promote the production of type I interferons (IFNs) and IFNγ [98], which is significant because activation of type I IFN expression can stimulate dendritic cells, leading to immune-mediated tumor rejection through CD8+ T-cell responses [99]. In addition to its immune-modulating effects, direct targeting of SAE has been found to inhibit tumor cell proliferation. Genome-wide RNAi screens have identified the genes encoding the SAE subunits (SAE1 and SAE2) as having the strongest synthetic lethal interactions with c-Myc [100]. Researchers indicated that inhibiting SAE activates the expression of the tumor suppressor miR-34, which targets the mRNA of c-Myc and other oncogenic pathways [101]. Furthermore, SAE is also a critical target for reducing cancer cell stemness [102].

The most successful approach in developing inhibitors of UbL E1 enzymes involves targeting their ATP-binding sites. MLN-4924 (or pevonedistat) is the first molecule resulting from this strategy, highly potent and specific in inhibiting E1 for UbL Nedd8, showing efficacy in treating acute myeloid leukemia [103,104]. The success of MLN-4924 has paved the way for the development of structurally related inhibitors specific to UbL E1 and SUMO E1. For example, the SUMO E1 inhibitor ML-792 displays selective cytotoxicity in c-Myc-overexpressing cells in preclinical studies [105]. ML-93, a derivative of ML-792, demonstrates strong selectivity in inhibiting SUMOylation through a similar mechanism of action in pancreatic cancer, which leads to G2/M phase arrest and promotes apoptosis [106]. Several natural products have also been identified that inhibit SUMO E1 activity, including ginkgolic acid [74,107,108], davidiin [109], tannic acid [110], and kerriamycin B [111]. However, the effectiveness and specificity of these natural compounds are limited, as indicated by their half maximal inhibitory concentration (IC50) values in the micromolar range and their broad range of targets. A newly discovered SAE inhibitor, COH000, targets a cysteine residue in the AAD without affecting the catalytic cysteine [112].

TAK-981, a first-in-class small molecule SUMOylation inhibitor, inhibits SAE similarly to ML-792 by forming an irreversible adduct with SUMO protein in an enzyme-catalyzed, ATP-dependent process. TAK-981 has been found to activate IFN1 signaling to promote antitumor immune responses and is currently undergoing phase I clinical trials [113]. Specifically, an in vivo study revealed that TAK-981 enhanced the proportions of activated CD8 T-cells and natural killer (NK) cells [114]. Furthermore, pretreatment with TAK-981 enhanced macrophage phagocytosis or NK cell cytotoxicity against CD20+ target cells in combination with the anti-CD20 antibody rituximab [115]. TAK-981 also provokes apoptosis and cell cycle arrest in acute myeloid leukemia [116].

The broad application of TAK-981 faces significant challenges, particularly regarding the potential toxicity risks linked to the widespread presence of SUMO modifications. Both systemic and local injections of TAK-981 can cause undesirable inflammation in normal tissues, resulting in adverse events such as diarrhea and ulceration [117]. Furthermore, its poor solubility and low bioavailability further restrict its clinical utility. Another challenge is the inability of systemic or local injections to provide sustained release. Notably, recent research proposes an injectable PDLLA-PEG-PDLLA (PLEL) nanocomposite hydrogel that incorporates self-assembled TAK-981 and BSA nanoparticles for localized treatment of residual tumors following iRFA. The sustained release of TAK-981 from this hydrogel inhibits the expansion of residual tumors and significantly stimulates dendritic cell and cytotoxic lymphocyte-mediated antitumor immune responses while ensuring biosafety. The development of an injectable drug delivery hydrogel holds promise for enhancing the efficacy and expanding the clinical applications of the SUMOylation inhibitor [118] (Table 1).

Several studies have reported that SENPs’ aberrant expression is associated with the development and progression of cancer [129,130,131]. SENPs are cysteine proteases with isopeptidase activity, play a crucial role in maintaining the balance between the pools of SUMOylated and unSUMOylated proteins and in SUMO recycling. MiR-145-mediated down-regulation of SENP1 induced quiescence of prostate cancer cells and reversed SENP1-promoted tumorigenesis in mice. This highlights the potential of miR-145 as a therapeutic molecule against cancer [132]. Thus, targeting SENPs could offer a promising strategy in cancer treatment by modulating the SUMOylation process.

Knockdown of SENP6 has been found to induce the radiosensitization of liver cancer cells, highlighting its potential to enhance the sensitivity of cancer cells to anticancer therapy [133]. In addition to impeding tumor growth, the silencing of SENP can play a crucial role in enhancing the effectiveness of anticancer treatments. The current research focuses on the discovery of SENP inhibitors with a specific emphasis on developing isoform-selective inhibitors, which poses a significant challenge. Various strategies have been employed in the design and development of SENP inhibitors, including non-peptidyl low-molecular-weight inhibitors [134], virtual-screening-assisted low-molecular-weight inhibitors [135], and natural compounds extracted from plants [124]. Despite these efforts, developing effective SENP inhibitors continues to be a complex and challenging task of cancer research.

Furthermore, a study identifies BMP8A, RGS8, and SERPIND1 as key biomarkers associated with SUMOylation in PTC, suggesting potential targets for therapeutic intervention and prognosis in PTC research [136]. Developing a nomogram based on the SUMOylation score could provide valuable insights for individualized treatment strategies in thyroid cancer. In lung adenocarcinoma, high SUMOylation scores correlate with poor prognosis [137], emphasizing the significance of understanding SUMOylation patterns in predicting patient outcomes. Their connection to immune response and drug sensitivity further underscores the importance of these biomarkers in PTC research. According to a recent study, protein SUMOylation levels in thyroid tumor tissues were higher than those in paired nontumor tissues, and the higher the SUMOylation in tumor tissues, the shorter the overall survival time of the patients, especially among males [138].

## 6. Conclusions and Future Perspective

The study of SUMOylation in thyroid cancer has illuminated its pivotal role in tumorigenesis and progression, opening up promising avenues for therapeutic intervention. Our comprehensive review underscores the dual nature of SUMOylation, highlighting its capacity to both promote and suppress oncogenic processes. This complexity suggests that nuanced strategies targeting SUMOylation could yield significant therapeutic benefits.

Key findings of this review emphasize the aberrant SUMOylation patterns in thyroid cancer, particularly in differentiated and anaplastic subtypes, and the potential to exploit these modifications for targeted therapies. The involvement of SUMOylation in crucial cellular processes—such as DNA repair, transcription regulation, and protein stability—underscores its broad impact on cancer biology and its potential as a therapeutic target. The deregulation of SUMO machinery components and the SUMOylation of specific proteins like CCDC6, PDGF-C, and TFAP2A offer novel insights into thyroid cancer pathogenesis and progression.

Future research should focus on elucidating the detailed molecular networks and pathways modulated by SUMOylation in thyroid cancer, considering the heterogeneity among different subtypes. Additionally, developing specific inhibitors of the SUMOylation pathway, including SUMO E1 enzyme inhibitors and SENP protease modulators, could provide innovative therapeutic options. It is crucial to integrate these strategies with existing treatment modalities to enhance efficacy and overcome resistance.

Furthermore, preclinical and clinical trials are necessary to validate the safety and effectiveness of targeting SUMOylation in thyroid cancer patients. Such trials will pave the way for personalized treatment approaches, taking into account individual SUMOylation profiles and tumor characteristics. By advancing our understanding of SUMOylation’s role in thyroid cancer, we can develop more precise and effective therapies, ultimately improving patient outcomes and quality of life. The potential of SUMOylation-based therapies to revolutionize the treatment landscape underscores the need for continued exploration and innovation in this promising field.

## Figures and Tables

**Figure 1 biomedicines-12-02408-f001:**
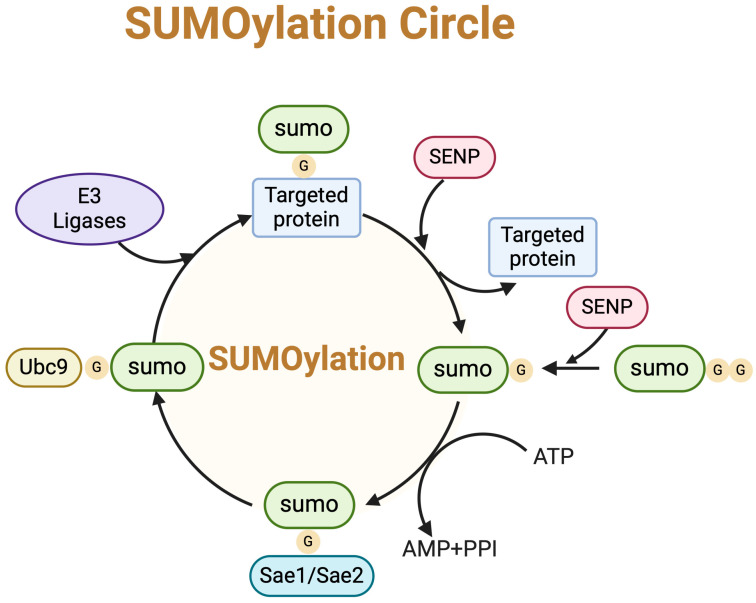
The visualization of SUMOylation circle. SUMO precursors are processed by Sentrin-specific protease 1 (SENP1), SENP2, and SENP5 to yield SUMO-GG, which is activated by SUMO E1 (SAE1/SAE2), transferred to SUMO E2 (UBC9), and assisted by a SUMO E3 ligase to conjugate to substrates. SUMO can be removed from SUMO-conjugated substrates through SENPs.

**Figure 2 biomedicines-12-02408-f002:**
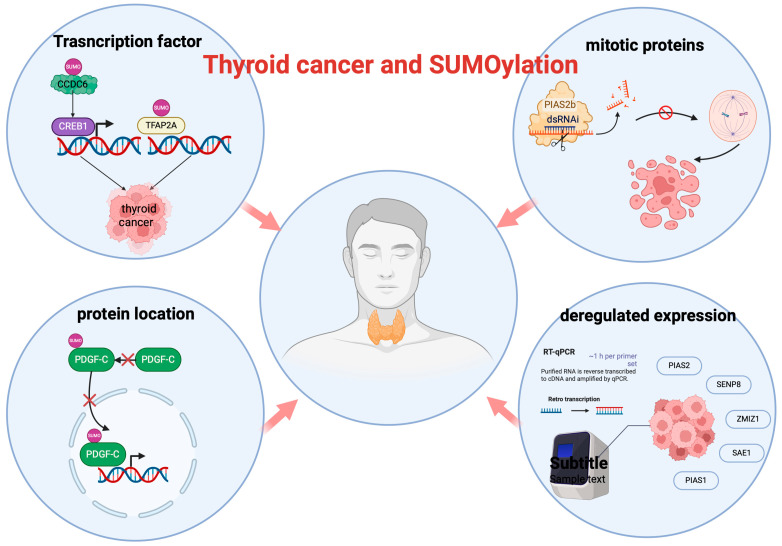
SUMOylation and Thyroid cancer.

**Table 1 biomedicines-12-02408-t001:** SUMOylation Inhibitors.

Inhibitor ^1^	Target	Cancer Type	Refs.
ZHAWOC8697	SENP1 and SENP2	-	[119]
Ursolic acid	SENP1	Hepatocellular carcinoma	[120]
SENP12-(4-Chlorophenyl)-2-oxoethyl 4-benzamidobenzoate derivatives	SENP1	Prostate cancer	[120]
1-[4-(N-benzylamino) phenyl]-3-phenylurea derivatives	SENP1	Cervical carcinoma	[120]
Gallic acid	SENP1	Colorectal cancer	[121,122]
BW467C60	SENP1	-	[123]
Triptolide	SENP1	Prostate cancer	[124]
Momordin Ic	SENP1	Acute myeloid leukemia, colon cancer, prostate cancer	[125]
Ginkgolic acid	E1	Gastric cancer, Breast cancer, Uveal melanoma	[126]
Anacardic acid	E1	Thyroid cancer, nonpromyelocytic acute myeloid leukemia, breast cancer, colon cancer, B-cell lymphoma	[126]
Kerriamycin B	E1	-	[111]
Davidiin	E1	Gastric cancer	[109]
Tannic acid	E1	-	[110]
compound 15	E1	-	[127]
COH000	E1	-	[112]
ML-792	E1	Hepatocellular carcinoma, pancreatic cancer	[105]
ML-93	E1	Pancreatic cancer	[106]
TAK-981	E1	Leukemia, acute myeloid, hepatocellular carcinoma, chronic lymphocytic leukemia, glioblastoma, pancreatic cancer, multiple myeloma	[128]

^1^ A plethora of natural and synthetic SUMOylation inhibitors have been identified.

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
