# Peer review of "The SUMO Family: Mechanisms and Implications in Thyroid Cancer Pathogenesis and Therapy"

_biomedicines, 2024, doi:10.3390/biomedicines12102408_

Round 1
Reviewer 1 Report
Comments and Suggestions for Authors
This review aims to evaluate the role of SUMO in the dysregulation of cellular processes such as protein localization, stability, and genome integrity. It explores the SUMO family, including its isoforms and catalytic cycle, emphasizing their importance in regulating key biological functions in thyroid cancer through SUMOylation. The authors also highlight the importance of developing effective SUMOylation inhibitors. The proposition of this review is interesting for developing a strategy for cancer treatment and prevention. However, several points need to be addressed:
1. The similarity index is 43%, which is unacceptable and must be reduced to below 20%.
2. The section on SUMOylation is poorly written and lacks a description of the mechanisms involved in this reaction.
3. SUMOylation involves key enzymes; it is important to mention them. Additionally, could inhibiting these enzymes be a viable strategy for tumor treatment?
4. An abstract figure would be beneficial to help readers better understand the mechanisms of SUMOylation and its role in disease development.
Author Response
Dear reviewer,
We are thankful for the positive responses. Following the suggestions, we thoroughly revised the manuscript and addressed the issues you mentioned. Revisions are marked using the “Track Changes” function. The corrections in the paper and the response are as follows(we always submitted the revised version without Track changes as PDF):
- The similarity index is 43%, which is unacceptable and must be reduced to below 20%.
Response: Thank you for bringing the similarity index to my attention. I apologize for the oversight. I have thoroughly revised the manuscript to address this issue.
- The section on SUMOylation is poorly written and lacks a description of the mechanisms involved in this reaction.
Response: Thank you for your valuable feedback regarding the section on SUMOylation. I have carefully revised the Introduction and the section on SUMO proteins and the SUMO catalytic cycle to provide a more detailed and clear explanation of the mechanisms involved in SUMOylation. The revised manuscript now includes a comprehensive description of th this modification's processes and key components. I hope these revisions address your concerns and enhance the overall clarity and quality of the manuscript.
- SUMOylation involves key enzymes; it is important to mention them. Additionally, could inhibiting these enzymes be a viable strategy for tumor treatment?
Response: Thank you for your insightful feedback regarding the key enzymes involved in SUMOylation and the potential therapeutic implications. In response, I have revised the section on SUMO proteins and the SUMO catalytic cycle to specifically mention and describe the key enzymes involved in this process. Additionally, I have included a table in the therapeutic section that outlines these enzymes and discusses the potential for targeting them as a strategy for tumor treatment. I believe these additions provide a clearer understanding of the enzymatic roles in SUMOylation and their relevance to therapeutic applications.
- An abstract figure would be beneficial to help readers better understand the mechanisms of SUMOylation and its role in disease development.
Response: Thank you for your valuable feedback and for taking the time to review our manuscript. We appreciate your suggestion to include an abstract figure to enhance the understanding of the mechanisms of SUMOylation and its role in disease development. However, considering the segmented nature of our discussion and the complexity of the mechanisms involved, we believe that a single figure might not effectively capture and convey all the essential aspects. We have aimed to provide detailed explanations and relevant figures within each section to support reader comprehension.We hope these elements collectively facilitate a thorough understanding. Nevertheless, we are open to further suggestions and willing to make adjustments where feasible.
Reviewer 2 Report
Comments and Suggestions for Authors
Thank you for giving me the opportunity to review the manuscript with the title: “The SUMO Family: Mechanisms and Implications in Thyroid Cancer Pathogenesis and Therapy”.
This manuscript discusses the role of Small Ubiquitin-like Modifier (SUMO) proteins in the pathogenesis and therapy of thyroid cancer. They may influence various cellular functions such as DNA repair, protein stability, and transcription regulation. Aberrant SUMOylation contributes to tumorigenesis by altering gene expression, promoting immune evasion, and supporting cancer progression. Targeting SUMOylation pathways, like with the drug TAK-981, shows promise for treating thyroid cancer. SUMO inhibitors could be used to halt tumour growth, metastasis, and improve the efficacy of immunotherapy. Various components of the SUMOylation system are deregulated in thyroid cancer, such as SENP8 and PIAS2. I believe the work brings a significant contribution to the field, is well organized and comprehensively described, is sound and not misleading. The references are well managed. The English language is fine, no objections from me.
Here are my suggestion for the improvement of the manuscript.
-Discuss more the term "SUMOylation" in order to be well understood.
-In the section” 2.2. SUMO catalytic cycle “for a better understanding figure 1 should be placed near the text where is mentioned.
-In section “4 SUMOylation and thyroid cancer” also Figure 2 should be near the text.
- A table with all the abbreviations mentioned in each section must be placed at the beginning of the section to make easier the understanding of all abbreviations for the non-specialists that want to read the paper. Tyroid cancer is frequent, and many MDs want to read and understand it better. Because this is a review it must be accessible to all types of MDs.
- In section “ 5. The potential application of SUMOylation in the treatment of thyroid cancer”, explain pointed the limitations of surgery and radiation therapy in order to better contextualize the potential of SUMO inhibitors.
I also remarked the lack of clinical data, of course there are preclinical models (cell lines, murine models) presented, but place more emphasis on clinical studies and trials that have used SUMO inhibitors in thyroid cancer. You cam even discuss more Case-Specific Data and focus on tyroid cancer subtypes. For example subtypes of thyroid cancer such as papillary, follicular, and anaplastic and how SUMO impacts them.
- The figures discussing the SUMOylation cycle and its role in thyroid cancer could benefit from more detailed diagrams explaining the complex processes.
-The manuscript could benefit from a more detailed discussion of the study’s limitations or challenges in current SUMOylation research. This would provide a balanced perspective
- Expand the conclusion to clearly emphasize the key findings and implications for future research
- Explicitly explain how future SUMO-based therapies could be integrated with current thyroid cancer treatments and their benefits for the treated patients.
- So, more detailed discussion of clinical relevance, combination therapies, and future directions for research may improve your discussion.

Author Response
Dear Reviewer,
We thank you for the positive assessment of our work. Following your suggestions, we thoroughly revised the manuscript and addressed the issues. Revisions to the manuscript are marked using the “Track Changes” function. The corrections in the paper and the response are as follows(we always submitted the revised version without Track changes as PDF):
- -Discuss more the term "SUMOylation" in order to be well understood..
Response: Thank you for your valuable feedback regarding the section on SUMOylation. I have carefully revised the Introduction and the section on SUMO proteins and the SUMO catalytic cycle to provide a more detailed and clear explanation of the mechanisms involved in SUMOylation.
- -In the section” 2.2. SUMO catalytic cycle “for a better understanding figure 1 should be placed near the text where is mentioned.
- -In section “4 SUMOylation and thyroid cancer” also Figure 2 should be near the text.
Response: Thank you for your valuable suggestion to include an abstract figure to enhance the understanding of the mechanisms of SUMOylation and its role in disease development. Placing the figure alongside the relevant text would be ideal for clarity. However, per the journal's submission guidelines, all figures must be placed at the end of the manuscript during submission. Please rest assured that the journal's editorial team will reposition the figure appropriately during the final layout stage. We appreciate your understanding and support.
- - A table with all the abbreviations mentioned in each section must be placed at the beginning of the section to make easier the understanding of all abbreviations for the non-specialists that want to read the paper. Tyroid cancer is frequent, and many MDs want to read and understand it better. Because this is a review it must be accessible to all types of MDs.
Response: Response: Thanks for the valuable suggestion. According to the journal's submission guidelines, We added the abbreviated terms’ interpretation as a new section at the end of the manuscript. The detailed contents are as follows:
Abbreviated terms
APM, Antigen processing and presentation machinery; ATC, Anaplastic thyroid carcinomas; cAMP, Cyclic AMP; CREB, cAMP-response element-binding protein; CTLs, Cytotoxic T cells; DDR, DNA damage response; DESI1, DeSUMOylating isopeptidase 1; DSBs, Double-strand breaks; DTC, Differentiated thyroid cancer; EMT, epithelial-mesenchymal transition; ESCC, Esophageal squamous cell carcinoma; HCC, Hepatocellular carcinoma; HEY1, Hairy/enhancer of split related with YRPW motif; HIF, Hypoxia-inducible factor; HK2, Hexokinase 2; HR, Homologous recombination; IC50, Half maximal inhibitory concentration; IFNs, Interferons; IGF-1R, Insulin-like growth factor 1 receptor; MTC, Medullary thyroid cancer; NHEJ, Non-homologous end joining; NK, Natural killer; PDGF-C, Platelet-derived growth factor-C; PD-L1, Programmed cell death protein-1 ligand; PPARγ, Peroxisome proliferators-activated receptors γ; PTC, Papillary thyroid cancer; PTC1, The RET/papillary thyroid carcinoma 1; PTM, Protein post-translational modification; PXR, Pregnane X receptor; RAI, Radioactive iodine; ROS, Reactive oxygen species; RTKs, Receptor tyrosine kinases; SAE1/SAE2, SUMO-activating enzyme subunit 1/2; SENPs, Sentrin/SUMO-specific proteases; SIM, SUMO-interacting motif; SUMO, Small ubiquitin-like modifier; SUMOi, Pharmacological inhibition of SUMOylation; SUMOylation, SUMO modification; TF, Transcription factor; TL, Thyroid lobectomy; TT, Total thyroidectomy; UbL,Ubiquitin-like modifiers.
- - In section “ 5. The potential application of SUMOylation in the treatment of thyroid cancer”, explain pointed the limitations of surgery and radiation therapy in order to better contextualize the potential of SUMO inhibitors..
Response: Thank you for your insightful feedback regarding the discussion of the potential application of SUMOylation in the treatment of thyroid cancer. We have revised the section to include a more detailed explanation of the limitations of surgery and radiation therapy. By highlighting these limitations, we aim to contextualize better the potential of SUMO inhibitors as a therapeutic approach.
- I also remarked the lack of clinical data, of course there are preclinical models (cell lines, murine models) presented, but place more emphasis on clinical studies and trials that have used SUMO inhibitors in thyroid cancer. You cam even discuss more Case-Specific Data and focus on tyroid cancer subtypes. For example subtypes of thyroid cancer such as papillary, follicular, and anaplastic and how SUMO impacts them.
Response: Thank you for your valuable feedback regarding the emphasis on clinical studies and trials using SUMO inhibitors in thyroid cancer. We acknowledge the current limitations in clinical data specifically for thyroid cancer. We have included additional discussions on the preclinical models to address this and highlighted the necessity for further clinical research. We also referenced studies from other cancer types where SUMO inhibitors have shown promise, to provide a broader context for their potential application in thyroid cancer.
1. Kim, H.S., et al., TAK-981, a SUMOylation inhibitor, suppresses AML growth immune-independently. Blood Adv, 2023. 7(13): p. 3155-3168.
2. Liu, J., et al., Targeting SUMOylation with an injectable nanocomposite hydrogel to optimize radiofrequency ablation therapy for hepatocellular carcinoma. J Nanobiotechnology, 2024. 22(1): p. 338.
3. WU Q, et al., Correlationship between total proteins SUMOylation and papillary thyroid carcinoma in males[J]. J Surg Concepts Pract, in press. Published
online, DOI:10.16139/j.1007⁃9610.2024.04.00.
- - The figures discussing the SUMOylation cycle and its role in thyroid cancer could benefit from more detailed diagrams explaining the complex processes.
Response: Thank you for your valuable feedback regarding the figures on the SUMOylation cycle and its role in thyroid cancer. We appreciate your suggestion to include more detailed diagrams. However, as this manuscript is an invited short review, the figures are intended to serve as simplified illustrations that complement the more comprehensive discussion in the text. We hope that the detailed explanations in the text adequately support the reader's understanding of these complex processes. We appreciate your understanding. - -The manuscript could benefit from a more detailed discussion of the study’s limitations or challenges in current SUMOylation research. This would provide a balanced perspective.
Response: Your feedback is greatly appreciated. We acknowledge the limitations of the current research on SUMOylation in thyroid cancer. To provide a more comprehensive and balanced perspective, we have included a detailed discussion in section 5.2, "Therapeutic Potential and Development of SUMOylation Inhibitors," where we address the challenges and limitations identified in SUMOylation research across other types of cancer. This discussion provides context for the application of SUMOylation in thyroid cancer treatment and highlights areas that require further exploration and resolution in current research. We believe these additions will enhance the overall quality of the manuscript.
9.- Expand the conclusion to clearly emphasize the key findings and implications for future research
Response: We thank you for pointing out this. Following the reviewer’s suggestion, we have given detailed key findings and implications for future research in the conclusion section.
10.- Explicitly explain how future SUMO-based therapies could be integrated with current thyroid cancer treatments and their benefits for the treated patients.
Response: We are thankful for your valuable suggestions. Currently, limited clinical studies specifically examine SUMO-based thyroid cancer therapies. However, based on existing preclinical research, SUMOylation-targeted treatments could potentially be applied in personalized therapy approaches. More information about this is discussed in the 5.2. Therapeutic potential and development of SUMOylation inhibitors part. Further research is needed to explore these possibilities. Here are a few references that may provide valuable insights:
- Liu, J., et al., Targeting SUMOylation with an injectable nanocomposite hydrogel to optimize radiofrequency ablation therapy for hepatocellular carcinoma. J Nanobiotechnology, 2024. 22(1): p. 338.
Reviewer 3 Report
Comments and Suggestions for Authors
1. The introduction need enhance by adding more details about thyroid cancer
2. The manuscript needs other studied about SUMOylation in different cancers
3. the manuscript need more organize
4. added more details about SUMOylation
Author Response
Dear reviewer,
We are highly obliged for your positive assessment of our work. We have thoroughly revised the manuscript and addressed the issues. Revisions to the manuscript has been marked up using the “Track Changes” function. The corrections in the paper and the response are as follows(we always submitted the revised version without Track changes as PDF):
- The introduction need enhance by adding more details about thyroid cancer.
Response: We thank you for pointing out this. Following the reviewer’s suggestion, We added more details about the thyroid cancer treatments in the introduction part.
- The manuscript needs other studied about SUMOylation in different cancers.
Response: We are thankful for your valuable suggestions. We appreciate the suggestion to include studies on SUMOylation in different cancers. We have summarized these studies in a table within our manuscript. While we agree that placing the table in the relevant text section could enhance readability, the journal's submission guidelines require all tables to be placed at the end of the manuscript. The journal's editorial team will reposition the table appropriately during the final editing process.
- the manuscript need more organize.
Response: Thank you for your suggestions.I have rephrased and restructured the relevant sections to enhance originality while maintaining the integrity and clarity of the content. I hope the revised version meets the necessary standards. Please let me know if there are any additional changes required
- added more details about SUMOylation
Response: Thank you for your valuable feedback regarding the section on SUMOylation. I have carefully revised the Introduction and the section on SUMO proteins and the SUMO catalytic cycle to provide a more detailed and clear explanation of the mechanisms involved in SUMOylation.